# *cis*-1,4 Selective Coordination Polymerization of 1,3-Butadiene and Copolymerization with Polar 2-(4-Methoxyphenyl)-1,3-butadiene by Acenaphthene-Based *α*-Diimine Cobalt Complexes Featuring Intra-Ligand π-π Stacking Interactions

**DOI:** 10.3390/polym13193329

**Published:** 2021-09-29

**Authors:** Beibei Wang, Heng Liu, Tao Tang, Xuequan Zhang

**Affiliations:** 1Changchun Institute of Applied Chemistry, Chinese Academy of Sciences, 5625 Renmin Street, Changchun 130022, China; bbwang@ciac.ac.cn (B.W.); ttang@ciac.ac.cn (T.T.); 2Changchun Institute of Applied Chemistry, University of Science and Technology of China, Hefei 230026, China; 3Key Laboratory of Rubber-Plastics, Ministry of Education/Shandong Provincial Key Laboratory of Rubber-Plastics, Qingdao University of Science & Technology, Qingdao 266042, China; xqzhang@qust.edu.cn

**Keywords:** *cis*-1,4 selective, 1,3-butadiene, *α*-diimine cobalt complexes, copolymerization, π-π stacking interactions

## Abstract

Highly *cis*-1,4 selective (up to 98%) coordination–insertion polymerization of 1,3-butadiene (BD) has been achieved herein using acenaphthene-based *α*-diimine cobalt complexes. Due to the presence of intra-ligand π-π stacking interactions, the complexes revealed high thermostability, affording polybutadiene products in high yields. Moreover, all of the obtained polymers possessed a relatively narrow molecular weight distribution as well as high molecular weight (up to 92.2 × 10^4^ Dalton). The molecular weights of the resultant polybutadienes could be finely tuned by varying polymerization parameters, including temperature, Al/Co ratio, etc. Moreover, the copolymerization of butadiene with polar monomer 2-(4-methoxyphenyl)-1,3-butadiene (2-MOPB) was also successfully realized to produce a type of polar *cis*-1,4 polybutadiene (*cis*-1,4 content: up to 98.1%) with a range of 2-MOPB content (0.46–1.83%). Water contact angle measurements indicated that the insertion of a polar monomer into a polymer chain could significantly improve the polymer’s surface property.

## 1. Introduction

Hydrocarbon polymers, which mainly include polyolefins and polydienes, have been widely used in modern human life because of their excellent chemical and physical properties in combination with their low cost, superior processability, and good recyclability [1,2,3]. However, due to the lack of polar functional groups, most hydrocarbon polymers exhibit very poor surface properties, such as low adhesive properties, low compatibility with polar fillers, etc., which greatly limits their applications in a broader scope [4,5,6]. Copolymerization with polar monomers to access functionalized hydrocarbon polymers is the most direct and economic strategy to solve the above issue; nevertheless, most catalytic systems have extremely high oxophilic active species that are able to be decomposed by heteroatoms (such as N, O, S, or other polar groups) due to their strong and irreversible coordination to the metal center. In 1995, Brookhart et al. disclosed a family of *α*-diimine late transition metal-based catalytic systems [7,8]. Owing to the low oxophilic nature of the metal centers and the properties of *α*-diimine coordination ligand that are able to be regulated exactly, a catalytic system that shows a high heteroatom tolerance and that can directly catalyze the copolymerization of olefins with polar comonomers, has created a new field aiming to access functionalized polyolefins, and colossal advances have been made this field in the past few years [9,10,11,12]. Similar to polar polyolefin materials, functionalized polydienes elastomers are also of great significance because the incorporated functional groups can greatly improve the compatibility between the polydienes matrix and reinforce polar fillers, such as carbon black, silica, etc., which therefore produces well-performing elastomers with good modulus, tensile strength, and other dynamic mechanical properties [13]. However, research n copolymerization with polar copolymers in this field o is very limited [14,15]. In the present research, a series copolymerizations of 1,3-butadiene with polar comonomer 2-(4-methoxyphenyl)-1,3-butadiene (2-MOPB) is conducted, and the corresponding catalytic performances and microstructures of the resulting copolymers are discussed.

Despite of the inherent advantage of late-transition metal complexes for the direct preparing of polar copolymers, they still currently suffer from some shortages, and one of them is the thermal stability of the active species. Currently, the specific high-temperature decomposition mechanism for late-transition metal catalysts is still unclear, yet one widely accepted explanation ascribes it to the C-H activation that occurs between the metal center and alkyl groups located on the ortho-position of the *N*-aryl fragment after it rotates to the coordination plane [16]. Therefore, suppressing *N*-aryl rotations is believed to be one of the most efficient strategies to enhance the thermal stability of these complexes, and in the past few years, various intra-ligand covalent and non-covalent interactions, e.g., hydrogen bonding, π,π-stacking, etc., have been introduced to ligand design, aiming to fasten the *N*-aryl moiety and to subsequently prohibit its rotation [17,18,19]. In the present research, ortho-substituted dibenzhydryl groups were introduced to the *N*-aryl moiety of *α*-diimine cobalt complexes, and surprisingly, it was found that one of the aryl groups in dibenzhydryl could form π,π-interaction with the acenaphthenyl backbone, which subsequently endows the complexes with excellent thermal stability. Detailed research discussing these findings will be presented in the following sections.

## 2. Materials and Methods

### 2.1. General Considerations and Materials

All reactions were conducted under a dry and oxygen-free argon atmosphere using Schlenk techniques or under a nitrogen atmosphere in an MBraun glovebox (MBraun, Garching bei München, Germany). Toluene for polymerization was dried with sodium and was distilled under the nitrogen atmosphere. Dichloromethane and diethyl ether were also dried with calcium hydride and were distilled under the nitrogen atmosphere. EASC (Aladdin, Shanghai, China) was purchased and then diluted in a 2 M toluene solution. Catalysts were dissolved in toluene and were added to the polymerization vessel with the injector. Other commercially available reagents were purchased and were used without purification. Ligands **L1–L4** were synthesized following the procedures reported upon in the literature [17]. The compound 2-MOPB was also prepared according to previous reports [11].

NMR spectra of the organic compounds were recorded on a Bruker AV400 (Bruker Optics, Ettlingen, Germany) (FT, 400 MHz for ^1^H; 100 MHz for ^13^C) spectrometer at room temperature with tetramethylsilane (TMS; *δ* = 0 ppm) as an internal reference, and then the NMR spectra of copolymers were measured at a room temperature using a Varian Unity 400 MHz spectrometer (Varian, Inc., Palo Alto, CA, USA). Polymer characteristics were investigated on a TOSOH HLC-8220 GPC instrument (Tosoh Asia Pte. Ltd., Tokyo, Japan) at 40 °C with polystyrene as a standard. Tetrahydrofuran (THF) was used as the eluent at a flow rate of 1.0 mL min^−1^. The polymer microstructure analysis was performed using a Nicolet Is10 FTIR (Thermo fisher, Waltham, MA, USA). Copolymer water contact angles were measured using a JY-PHB contact angle meter (YouTe, Chengde, China).

The crystal of **Co1** was obtained by hexane diffusion to its saturated dichloromethane solution, while the crystal of **L2** precipitated from its saturated dichloromethane/hexane mixed solution at room temperature. All of the crystal performances took place in the nitrogen atmosphere, and data collection took place at a low temperature on a Bruker SMART APEX diffractometer (Bruker Optics, Germany) with a CCD area detector using graphite monochromated Mo Kα radiation (*λ* = 0.71073 Å). The raw frame data were processed using SAINT (Bruker Optics, Germany) and SADABS (Bruker Optics, Germany) to yield the reflection data file. The crystal class characteristics and unit cell parameters were performed with the SMART program package. The structures were determined using the SHELXTL program. The hydrogen atoms were located at the calculated positions and were included in the structure calculation without further refinement of the parameters. CCDC numbers for **Co1** and **L1**: 2095433-2095434.

### 2.2. Typical Procedure for Butadiene Homopolymerization and Copolymerization

Polymerizations of 1,3-butadiene were performed by introducing butadiene (0.1 g/mL in toluene) and EASC solution into 20 mL glass flasks equipped with a magnetic stirrer to mix the solution. At the setting polymerization temperature, a cobalt complex toluene solution was injected. Then, the polymerization was initiated and performed for a designated amount of time. After the polymerization was finished, acidified MeOH (HCl, 10 wt%) and antioxidant were added, and the polymer was collected and purified by pouring ethanol (150 mL) into a flask to precipitate. The polymer was dried at 40 °C in a vacuum oven to a constant weight. For the copolymerization of 2-MOPB and butadiene, the same procedure was performed except for that the mixed monomer of the evaluated amount of 2-MOPB and butadiene was added to the glass flasks for polymerization.

### 2.3. Synthesis of Complexes (**Co1**–**Co4**)

#### 2.3.1. Synthesis of Complex **Co1**

To a mixture of **L1** (240 mg, 0.32 mmol) and CoCl_2_ (42 mg, 0.32 mmol), dichloromethane (20 mL) was added at room temperature. Then, the mixture was stirred overnight at room temperature. The solvent was removed until 2 mL were left. An amount of 50 mL hexane was added while the mixture was being stirred. The mixture was filtered, and the residue was washed with hexane (3 × 10 mL) and was dried under vacuum; the pure target product (197 mg, 0.22 mmol) was collected as a brown solid at a 69.9% yield. FTIR (KBr, cm^−1^): 3057(w), 3024(w), 2958(w), 2920(w), 2853(w), 1652(*v*_C=N_, s), 1466(m), 1450(m), 1383(w), 1355(w), 1317(w), 1204(w), 1157(w), 1095(w), 1025(w), 862(m). MS (ESI): m/z calcd. for C_56_H_48_Cl_2_CoN_2_, [M]^+^ 878.83, found [M-Cl]^+^ 842.3.

#### 2.3.2. Synthesis of Complex **Co2**

The procedure was similar to the synthesis of **Co2**. Yield: 215 mg, 0.23 mmol, (yield: 73.0%). FTIR (KBr, cm^−1^): 3058(w), 3025(w), 2959(w), 2920(w), 2853(w), 1647(*v*_C=N_, s), 1469(m), 1452(m), 1383(w), 1357(w), 1290(w), 1174(w), 1089(w), 1031(w), 1016 (w), 862(m). MS (ESI): m/z calcd. For C_54_H_42_Cl_4_CoN_2_, [M]^+^ 919.67, found [M-Cl]^+^ 884.2.

#### 2.3.3. Synthesis of Complex **Co3**

The procedure was similar to the synthesis of **Co3**. Yield: 149 mg, 0.16 mmol, (yield: 51.2%). FTIR (KBr, cm^−1^): 3059(w), 3026(w), 2959(w), 2921(w), 2852(w), 1648(*v*_C=N_, s), 1452(m), 1440 (m), 1383(w), 1204(w), 1177(w), 1092(w), 1033(w), 863(m). MS (ESI): m/z calcd. for C_56_H_48_Cl_2_CoN_2_O_2_, [M]^+^ 910.83, found [M-Cl]^+^ 874.2.

#### 2.3.4. Synthesis of Complex **Co4**

The procedure was similar to the synthesis of **Co4**. Yield: 235 mg, 0.25 mmol, (yield: 77.3%). FTIR (KBr, cm^−1^): 3054(w), 3025(w), 2956(w), 2920(w), 2854(w), 1650(*v*_C=N_, s), 1451(m), 1437(m), 1383(w), 1364(w), 1318(w), 1203(w), 1076(w), 1033(w), 862(m). MS (ESI): m/z calcd. for C_62_H_48_Cl_2_CoN_2_, [M]^+^ 950.9, found [M-Cl]^+^ 915.3.

## 3. Results

### 3.1. Synthesis and Characterization of the Cobalt Complexes

Ligands **L1–L4** were synthesized by the condensation reaction between ortho-substituted aniline and acenaphthene according to the reported method [20,21]. Their corresponding cobalt complexes **Co1****–Co4** could subsequently be accessed in high yields by directly treating **L1–L4** with a stoichiometric amount of CoCl_2_ (Figure 1) in dichloromethane and with subsequent precipitation by diethyl ether. All of the complexes **Co1–Co4** were well characterized by FTIR and mass spectra. In the FTIR spectra, the stretching vibration absorptions of the C=N groups in complexes **Co1–Co4** appeared at 1647.9–1652.7 cm^−1^, which was shifted slightly compared with their counterparts in the free ligands, indicating good coordination between the *α*-diimine nitrogen atoms and the metal cobalt. Single crystal structures of ligand **L2** and complex **Co1** were further characterized by single crystal X-ray crystallographic analysis, and their structures are shown in Figure 1 and Figure 2. As expected, a distorted tetrahedron geometry was observed for complex **Co1**, and the bond angles and bond distances in **L2** and **Co1** are typical for previously reported *α*-diimine Co(II) complexes [22,23,24].

For the ligand **L2**, both imine groups revealed *E,E*-configurations, which were different from some of the previous literature, which stated that *E,Z*-configurations were adopted [25,26,27,28]. Such a difference was probably due to the intra-ligand π,π-interactions that were found between the acenaphthenyl plane and the phenyl rings on the dibenzhydryl groups, which favored the *E,E*-configuration as the more thermodynamically stable state. As seen from the illustrative structure shown in Figure 1 (right), the two phenyl rings on the dibenzhydryl groups (colored into red) connected to the ortho-position of the aniline fragments were nearly parallel with the acenaphthenyl plane (colored into blue), revealing dihedral angles of only 25.85° and 21.89°, respectively. The distances from the center of the two phenyl rings to the acenaphthenyl plane are also very short, with values of 3.386 Å and 3.388 Å, respectively. The nearly parallel structure as well as the short distances clearly indicate the presence of intra-ligand π,π-interactions.

Such a π,π-interaction can be also clearly observed in the complex **Co1**. Similar to **L2**, the distances from the centers of the two phenyl rings (colored into red) on the dibenzhydryl groups to the acenaphthenyl plane (colored into blue) are also very short, producing values of 3.408 Å (due to the symmetric structure, the two distances are identical) (Figure 2). The dihedral angles between the two phenyls and the acenaphthenyl plane are even smaller, producing values of only 11.26°. Such a smaller dihedral angle indicates a stronger π,π-interaction in the complex than the one in the ligand. This result makes sense when considering the electrons flowing from the acenaphthenyl group to the imine groups after coordination with the cobalt metal, which renders acenaphthenyl more electron-deficient and therefore produced a stronger interaction with the electron-rich phenyl rings on the dibenzhydryl substituents [18,19].

### 3.2. 1,3-Butadiene Homopolymerization

Cobalt complexes have previously been explored as efficient precatalysts for 1,3-diene polymerization [29,30,31,32]; nevertheless, most of them suffer from poor thermostabilities, i.e., at high temperatures (generally higher than 50 °C), the active species deactivated fast and produced polybutadiene products at rather low yields. Until now, no relative literature has put forward a specific reason for this; however, inspired by late-transition metal-mediated ethylene polymerization, the active species could be deactivated by C-H activation between the metal center and the ortho-substituted alkyl groups on the aniline fragments [16]. Suppressing such C-H activation through the rotation restrain of *N*-aryls has previously been proven as an effective strategy to enhance the thermostabilities of late-transition metal-based catalysts [17,33]. For the present cobalt complexes, due to the presence of intra-ligand π,π-interactions, such *N*-aryl rotations and thereby C-H activations could be also suppressed, resulting in the anticipation of improved thermostabilities. Based on the above consideration, the present cobalt complexes were subsequently explored for 1,3-butadiene polymerization at various temperatures.

All of the 1,3-butadiene homopolymerizations employed ethylaluminum sesquichloride (EASC) as a cocatalyst because in our previous reports, EASC was determined to be one of the most efficient cocatalysts. Before exploring the thermal catalytic performances, the cocatalyst EASC/Co ratio was first optimized. As per the data shown in Table 1, with the increment of EASC loading, gradually increasing catalytic activities were witnessed for **Co1** mediated systems, which was consistent with previous reports. Moreover, monotonously decreasing molecular weights of the resultant polybutadienes were revealed, indicating the facilitated polybutadienyl chain transfer reactions from the propagating cobalt active species to aluminum alkyl compounds as well as increased activation and formation of catalytic species when the EASC equivalents increased. At EASC/Co = 300, the resulting polymer possesses a similar molecular weight to a commercial polybutadiene counterpart that has a Mooney viscosity of approximately 45; therefore, this condition was applied to subsequent studies. It is worthy of noting that the *cis*-1,4- contents of the polybutadienes almost remained completely unchanged when varying the Al/Co ratios, indicating their subtle influences on the regioselectivity of the active species.

The thermal stabilities of the complexes **Co1–Co4** were next investigated systematically by conducting polymerization at various temperatures. As per the data summarized in Table 2, their thermal catalytic performances were highly dependent on the ligand structures. For complex **Co1**, its catalytic activity increased dramatically from 0.6 × 10^4.1^ to 6.4 × 10^4^ g PBD (mol of Co)^−1^ h^−1^ when the temperature was elevated from 0 °C to 50 °C; nevertheless, further temperature increases led to a completely deactivated system, suggesting the poor stability of the corresponding active species. Similar results were observed for complexes **Co2** and **Co3**: their highest catalytic activities were achieved at 30 °C; however, elevating the temperature resulted in quickly deactivated systems, and at 70 °C, no polymer products were obtained. In contrast to **Co1–Co3**, outstanding high catalytic efficiency as well as high thermostability were revealed from **Co4**-mediated polymerizations. As per the data shown in Table 1 and Figure 3, all of the 1,3-butadiene monomers were converted to polymer products when conducting the polymerization at 30 °C with a catalytic activity of 8.3 × 10^4^ g PBD (mol of Co)^−1^ h^−1^; increasing the temperature to 50 °C had little influence on the polymer yields, indicating the much better thermostability of the active species than the other three analogs. To our satisfaction, **Co4** also afforded high polymer yields at high temperatures between 70 and 90 °C, at which point, most of the late transition metal complexes had been deactivated. These outstanding catalytic efficiencies of **Co4** at elevated temperatures were probably due to the enhanced π,π-interaction between the electron-rich naphthyl group with an acenaphthenyl backbone that could suppress the rotation of the *N*-aryls at a high temperature; for the other three complexes, however, the π,π-interaction interaction was comparatively weaker, and the stacked structure would have decomposed at high temperatures, which therefore could not guarantee the thermostability of the active species. Another thing worthy of note is that quite a high reaction rate was revealed from the **Co4**-mediated systems; when implementing polymerization at 50 °C, the monomer could be fully converted in only 10 min, demonstrating high catalytic activity of 99.6 × 10^4^ g PBD (mol of Co)^−1^ h^−1^.

Temperature also imposed a significant influence on the microstructures and molecular weights of the resultant polymers. For **Co4**-mediated polymerizations, a monotonous *cis*-1,4- content decrease of from 98.1% to 86.9% occurred when the temperature increased from 0 °C to 90 °C. Such a decrease can be ascribed to the facilitated *anti*–*syn* isomerization of the terminal π-allylic propagating unit through the π–σ-rearrangement at high temperatures, leading to decreased *cis*-1,4- content and simultaneously increased *trans*-1,4- units (Figure 4). Regarding the molecular weight of the resultant polymers, monotonously decreased *M*_n_s due to facilitated chain transfers at high temperatures were revealed.

### 3.3. Copolymerization of 2-(4-Methoxyphenyl)-1,3-butadiene with 1,3-Butadiene

Considering the high catalytic efficiency of complex **Co4**, the copolymerization of 1,3-butadiene with polar comonomer 2-(4-methoxyphenyl)-1,3-butadiene (2-MOPB) was conducted next to improve the surface properties of polybutadiene elastomers, and the results were summarized in Table 2. It was determined that although **Co4** could successfully catalyze the copolymerizations, the polar comonomer incorporation rate was very low, even when 50 equivalents of 2-MOPB to **Co4** were introduced. Under these conditions, the incorporated contents are shown to be in the range of 0.46–1.83%. The ^1^H and ^13^C NMR spectra of poly(2-MOPB-*co*-BD) are shown in Figure 5. Due to the relatively low incorporation of comonomer in the polymer, it can be reasonably predicted that the 2-MOPB unification was always isolated in the polymer chain. For the ^1^H NMR of the resultant polar copolymer, the resonances at *δ* 2.22 and 2.49 ppm were assigned to the newly generated methylene groups directly bonding to the double bond, which corresponded to the resonances at *δ* 26.97 and 29.70 ppm in the ^13^C NMR spectrum, respectively, while the resonances at *δ* 5.58 ppm corresponded to the double bond groups generated after 1,4 insertion. Moreover, ^1^H NMR produced the singlet at 3.79 ppm, which can be ascribed to the methoxy group that can also be observed at 55.25 ppm in the ^13^C NMR spectrum. The signals at 6.84 ppm corresponded to the aryl protons of 2-MOPB. This is consistent with previous reports [11]. When it comes to the other aryl hydrogens (H_3_), they may be overlapped by the *d*-chloroform resonances. Further analysis on the ^13^C NMR spectrum revealed that the signals at *δ* 161.20, 127.42, and 113.55 ppm corresponded to the aryl carbons of 2-MOPB. Moreover, some of the other groups were hardly assigned due to the low incorporation. According to these results, the copolymerization of 2-MOPB and butadiene was achieved successfully. Due to the poisoning of the active species by the methoxyl group in the comonomer, gradually decreased polymer yields as well as the molecular weights of the resultant polybutadienes were demonstrated. The static water contact angle (WCA) was tested for the copolymers to determine the changes in the surface properties. As per the results shown in Figure 6, the WCA of P(2-MOPB-*co*-BD) decreased from 95.6° to 89.1° when the 2-MOPB content increased from 0.46% to 1.83%, which was much lower than the value of 102.9° in PBD, which is indicative of the much-improved surface properties.

## 4. Conclusions

A new series of acenaphthene-based α-diimine cobalt complexes were synthesized that can polymerize 1,3-butadiene to access highly *cis*-1,4 selective (up to 98%) polybutadiene in all conditions. Due to the formed intra-ligand π-π stacking interactions, the complexes performed high catalytic activities at elevated temperature, and for complex Co4, the activities were maintained at 7.0 × 10^4^ g PBD (mol of Co)^−1^ h^−1^, which had not been reported before. Furthermore, all of the obtained polymers possessed a relatively narrow molecular weight distribution as well as high molecular weight (up to 92.2 × 10^4^ Dalton). The complexes were also able to promote the copolymerization of 1,3-butadiene with polar 2-MOPB, directly producing functionalized polymers, although their incorporation content was limited. After introducing polar 2-MOPB into the backbone, the surface properties of the polymers significantly improved, and greatly decreased water contact angles were revealed.

## Data Availability

The data presented in this study are available on request from the corresponding author.

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
