# Peer review of "cis-1,4 Selective Coordination Polymerization of 1,3-Butadiene and Copolymerization with Polar 2-(4-Methoxyphenyl)-1,3-butadiene by Acenaphthene-Based α-Diimine Cobalt Complexes Featuring Intra-Ligand π-π Stacking Interactions"

_polymers, 2021, doi:10.3390/polym13193329_

Round 1
Reviewer 1 Report
The article 'Cis-1,4 Selective Coordination Polymerization of 1,3-Butadiene and Copolymerization with Polar 2-(4-Methoxyphenyl)-1,3-butadiene by Acenaphthene-Based α-Diimine Cobalt Complexes Featuring Intra-ligand π-π Stacking Interactions' describes regiospecific polymerization of 1,3-butadiene using Co (II) diimine complexes. In general, this type of the catalysts is known, hiowever, the authors had represented a refreshing development in this field of coordination catalysis. The addition of minimal amounts of the polar diene copolymer, 2-(4-methoxyphenyl)-1,3-butadiene, resulted in significant improvement of the properties of poly(diene) in terms of surface property (lower water contact angle).
The scientific level of the article meet the high requirements of the Polymers journal.
However, the quality of the manuscript can be improved, bearing in mind the comments below.
Line 30 and below – the references should be presented in square brackets
Line 67 – stability instead of stabilities
Line 73 – purification of diethyl ether over CaH2 seems strange
Line 86 – 'constitutions' (???) – machine transtalion?
The list of corrections could be expanded, however, that would be full-fledged scientific editing. The text of the manusctipt contains a large number of similar rough edges, and needs substantial editing.
However, I have no profonud observations which would hinder the acceptance of this article.
Reviewer 2 Report
The paper of Liu et al. reports the synthesis of 4 Co-based catalytic precursors and their use in the homo-polymerization of 1,3-butadiene and co-polymerization of 1,3-butadiene with the polar monomer 2-(4-methoxyphenyl)-1,3-butadiene.
My comments are reported below:
- The authors hypothesize that the observed decrease of molecular weights of the resultant polybutadienes is due to the facilitated polybutadienyl chain transfer reactions from the propagating cobalt active species to aluminium alkyl compounds when increasing the EASC equivalents. The reason should be also attributed to the increase of the formed catalytic active species that could occur increasing the EASC equivalents.
- 13C NMR spectra of the copolymers should be also reported in the paper. 13C NMR should allow to precisely establish the copolymer chemo- regio- and stereo-chemistry.
- Finally, some English and typing errors are present in the manuscript. They should be corrected.
